# Associations of Electronic Device Use and Physical Activity with Headaches in Saudi Medical Students

**DOI:** 10.3390/medicina60020299

**Published:** 2024-02-09

**Authors:** Ahmad Y. Alqassim, Abdullah A. Alharbi, Mohammed A. Muaddi, Anwar M. Makeen, Waleed H. Shuayri, Abdelelah M. Safhi, Abdulrahman Y. Alfifa, Idris H. Samily, Nawaf A. Darbashi, Mohammed A. Otayn, Abdulaziz Y. Moafa, Ahmed M. Wafi, Mohamed Salih Mahfouz

**Affiliations:** 1Family and Community Medicine Department, Faculty of Medicine, Jazan University, Jazan 45142, Saudi Arabia; aaalharbi@jazanu.edu.sa (A.A.A.); mothman@jazanu.edu.sa (M.A.M.); amakeen@jazanu.edu.sa (A.M.M.); mm.mahfouz@gmail.com (M.S.M.); 2Faculty of Medicine, Jazan University, Jazan 45142, Saudi Arabia; s.wallid15@gmail.com (W.H.S.); s.safhi@gmail.com (A.M.S.); abdurrahman20304050@gmail.com (A.Y.A.); endreas14@gmail.com (I.H.S.); n8w8f1422@gmail.com (N.A.D.); mohammed1otain@gmail.com (M.A.O.); newten02000@gmail.com (A.Y.M.); 3Physiology Department, Faculty of Medicine, Jazan University, Jazan 45142, Saudi Arabia; amwafi@jazanu.edu.sa

**Keywords:** headache, migraine, medical student, exercise, technology, lifestyle, device use, physical activity, Saudi Arabia

## Abstract

*Background and Objectives:* Primary headaches are highly prevalent among medical students, negatively impacting their health and academic performance. Excessive electronic device use has been implicated as a risk factor, in contrast to physical activity, which may be a protective factor; however, comprehensive data are lacking, especially for Saudi medical trainees. This study aims to investigate the associations between device use, exercise, and headaches among Saudi medical students. *Materials and Methods:* In this cross-sectional study, 504 medical students at Jazan University completed an online survey collecting sociodemographic factors, headache characteristics/triggers, electronic device habits, exercise frequency, and headache impacts. Descriptive analyses summarized sample characteristics. Logistic regression identified predictors of 12-month headache prevalence. *Results:* Overall, 83% reported experiencing headaches in the past year. High headache prevalence was observed among females (86.6%) and third-year students (88.3%). Using electronic devices ≥4 h daily was associated with higher adjusted odds of headaches (OR 13.89, 95% CI 1.96–98.54) compared to ≤1 h daily. Low physical activity (exercising 1 day vs. 7 days a week) also increased headache odds (OR 3.89, 95% CI 1.61–9.42). Headaches impairing productivity (OR 4.39, 95% CI 2.28–8.45) and exacerbated by exercise (OR 10.37, 95% CI 2.02–53.35) were further associated with headache susceptibility. *Conclusions*: Excessive electronic device use and physical inactivity appear to be modifiable risk factors for frequent headaches in Saudi medical students. Multifaceted interventions incorporating education campaigns, skills training, and support services focused on promoting responsible technology habits, and regular exercise may help mitigate headaches in this population. Robust longitudinal studies and trials are warranted to establish causal mechanisms between lifestyle factors and headaches among medical undergraduates.

## 1. Introduction

Primary headaches such as migraine and tension-type headaches are distinct from secondary headaches attributed to underlying medical illness, head injury, or substance use [1,2]. Primary headaches, particularly migraine and tension-type headaches, are highly prevalent among medical students worldwide, with rates from 23% to 88% reported across studies [2,3,4,5]. These headaches can impair academic performance, decrease productivity, and negatively impact students’ quality of life [6,7]. The excessive use of electronic devices has been identified as a potentially modifiable risk factor for primary headache disorders in this population [8,9,10,11]. With the increasing integration of technology in medical education [12], Saudi medical students may be particularly susceptible to device-associated headaches [9]. However, few studies have comprehensively investigated headaches in relation to multiple dimensions like patterns, triggers, disability, and knowledge gaps among Saudi medical students specifically.

This is concerning because the unique stresses and demands of medical school may predispose students to headaches triggered or exacerbated by prolonged electronic device use for their studies [9,13]. Country-specific data evaluating these multidimensional associations are needed to guide context-appropriate interventions to alleviate headaches in Saudi medical students. Although lifestyle modifications like limiting device use and increasing physical activity have been suggested to reduce headache burden [14,15], compliance remains suboptimal partially due to knowledge deficits regarding effective headache prevention and self-management [16].

Several behavioral and lifestyle approaches have been investigated for migraine prevention and management, including stress management, dietary modifications, sleep hygiene, and increased physical activity [17]. In particular, growing evidence supports exercise as an effective intervention for reducing migraine frequency, pain, and disability [18]. For instance, according to a recent clinical practice guideline, aerobic exercise and yoga had the strongest grade B recommendation for decreasing migraine symptoms and attack frequency [19]. The guideline also provided specific exercise prescription recommendations based on the parameters used in clinical trials. However, compliance with lifestyle modifications remains suboptimal, highlighting the need for multidimensional interventions incorporating exercise training with education, self-management skills, and the promotion of behavioral change.

A number of studies have reported high headache prevalence among adolescents worldwide. For instance, a large Brazilian study found that 80.6% of students aged 14–19 years experienced headaches, with migraine (19.3%) and tension-type headaches (17.9%) being the most common primary types [20].

Therefore, this cross-sectional study aims to address this gap by thoroughly investigating electronic device use patterns, headache characteristics, disability, triggers, and knowledge in Saudi medical students. We hypothesize that excessive device use will be independently associated with increased headache prevalence, severity, and related disability. Secondly, we predict that students will exhibit limited knowledge regarding evidence-based prevention strategies and self-management of headaches. By elucidating modifiable correlates of headache susceptibility and disability concurrently with knowledge gaps, we intend to provide a reference for impactful educational initiatives, lifestyle recommendations, and self-care training to mitigate headaches and enhance productivity and well-being among this vulnerable population.

## 2. Materials and Methods

### 2.1. Study Area, Design, and Population

This cross-sectional study was conducted between September 2022 and April 2023 among undergraduate medical students at Jazan University, Saudi Arabia. The target population comprised all male and female medical students in their second through sixth year of study at the College of Medicine, Jazan University during the 2022–2023 academic year. Based on enrollment records, the sampling frame included 897 registered medical students. We aimed to recruit this full cohort using consecutive sampling in order to maximize sample size and statistical power. Students from any medical year willing to participate were eligible for inclusion. The only exclusion criterion was incomplete survey responses with more than 20% missing data. Medical students were targeted specifically because their intensive studying and electronic device use for educational purposes may predispose them to headaches, making this group appropriate to address the study’s aims.

### 2.2. Sampling Method

All 897 registered medical students were invited to participate in this study through an online self-administered questionnaire using Google Forms. The survey link was distributed via social media platforms and messaging apps popular among students at Jazan University. To encourage participation, reminders were sent in several successive rounds over a 2-month recruitment period from December 2022 to January 2023. Of the 897 students contacted, 518 completed the questionnaire (response rate of 56.2%). An additional 14 responses were excluded due to excessive missing data, leaving 504 responses for analysis. We originally aimed to recruit the full sampling frame to maximize sample size and statistical power.

### 2.3. Data Collection Tools

Data were collected using a structured self-administered online questionnaire developed by translating and adapting items from three previously validated surveys [16,21,22]. The initial English version was translated to Arabic by two independent bilingual translators and then back-translated to English to ensure accuracy. A linguist fluent in Arabic reviewed and edited the questionnaire for grammar and clarity. The final 53-item Arabic questionnaire contained five sections. The first section assessed sociodemographics: age, gender, marital status, academic year, grade point average, height, weight, parental education, income, and region. The second section evaluated headache patterns (e.g., frequency and severity) and electronic device use habits (e.g., type, duration, and frequency). The third section included 10 items assessing knowledge of headache symptoms, triggers, and treatments. The fourth section had three items on electronic device use and headache recurrence. The fifth section contained five items related to headaches’ effects on productivity and physical activity. Answer formats were multiple choice and Likert-type scales. Survey completion time was approximately 15 min. The questionnaire was anonymous and voluntary.

### 2.4. Data Analysis

Data analysis was conducted using SPSS version 26.0. Descriptive statistics including means, standard deviations, frequencies, and percentages were used to summarize sociodemographic variables and questionnaire responses. The primary outcome was self-reported headaches in the past 12 months (yes/no). The main exposure variable was daily electronic device usage (>4 h vs. ≤4 h). Bivariate analyses using chi-square tests were performed to assess univariate associations between device usage, sociodemographic factors, and headache prevalence. Multivariable logistic regression was used to identify independent predictors of primary headaches, adjusting for potential confounders including age, gender, academic year, and body mass index. Associations were expressed as adjusted odds ratios (aORs) with 95% confidence intervals (CIs). Secondary outcomes included headache frequency, severity, impacts on productivity, and physical activity. Bivariate and multivariable linear regression analyses were conducted for these continuous outcomes. For all analyses, a two-sided *p* < 0.05 was considered statistically significant. Missing data were minimal for most variables (<5%) and excluded pairwise in regression models.

## 3. Results

This cross-sectional study included 504 medical students at Jazan University, as shown in Table 1. The sample was predominantly male (54.2%, *n* = 273) and single (96.4%, *n* = 486), with a mean age of 21.5 years (SD 2.1). Overall, 82.5% (*n* = 414) reported experiencing headaches within the past year. The 12-month headache prevalence was significantly higher among females (86.6%, *n* = 200) compared to males (78.4%, *n* = 214) based on chi-square analysis (*p* = 0.017). By academic year, third-year students exhibited the highest headache prevalence at 88.3% (*n* = 98), while medical interns had the lowest at 74.4% (*n* = 29), although differences were not statistically significant. Students with moderate academic performance (CGPA 4.01–4.50) had lower headache prevalence (75.4%, *n* = 86) versus their higher-achieving peers. Overweight and obese students showed elevated headache rates, around 85%, compared to 81.4% for normal weight and 78% for underweight students, though BMI category differences were not significant. Mother’s education level showed a significant association, with lower headache prevalence among students whose mothers had graduate degrees (61.5%, *n* = 8) versus basic education (85.2%, *n* = 115, *p* = 0.025). These results identify subgroups with higher headache susceptibility, highlighting opportunities to target preventive interventions.

Table 2 shows the headache patterns and characteristics among the 504 participants stratified by gender. In the past month, the majority experienced headaches for 2–5 days (44.1%) with a typical headache duration of 31 min to 1 h (23.8%). Approximately half of males (51.6%) and females (51.7%) used 1–2 pills per headache episode; however, females used medications on more days in the past month (*p* = 0.016). The most common location was the forehead (42.6%), with the pain most often described as pressing/squeezing (41.1%). Light exposure (36.9%) and noise (32.5%) were the most frequently reported triggers. Females were more likely to experience light sensitivity (43.6% vs. 30.8%, *p* = 0.019), nausea (30.2% vs. 12.2%, *p* = 0.001), and vomiting (7.4% vs. 3.2%, *p* = 0.049) compared to males. Among the diagnosed participants (4.2%), migraines were most prevalent (33.3%). Figure 1 displays the coping strategies utilized by students during headache episodes. The most commonly reported technique was sleeping, endorsed by 90.9% of participants. Other frequently used approaches included practicing relaxation techniques (88%), resting in a quiet room (83.7%), ceasing electronic device use (72.7%), lying down in a dark room (67.2%), and drinking water (61.8%). Using hot or cold compresses was the least common method, reported by only 31.2% of students.

Table 3 presents the analysis results of the associations between electronic device usage and 12-month headache prevalence among the 504 medical students. Overall, 82.8% of device users reported headaches, compared to 44.4% of non-users (*p* = 0.003). The most commonly used devices were smartphones (78.2%) and tablets (86.0%). Notably, students using their primary device for over 4 h daily had higher headache prevalence (83.4%) versus those using their primary device for 1–2 h daily (61.9%, *p* = 0.006). This suggests excessive device time may be linked to headache susceptibility. However, no significant difference was observed between students using devices continuously (84.0%) versus intermittently (78.6%, *p* = 0.135).

Table 4 presents the analysis results of the relationship between physical activity and headache prevalence in the past year. Overall, 79.7% of students who exercised reported headaches, compared to 83.9% of non-exercisers, although this difference was not significant (*p* = 0.226). However, students who exercised regularly showed lower headache prevalence (74.7%) versus non-regular exercisers (83.6%, *p* = 0.053). Interestingly, exercise frequency and duration were significantly associated with headaches. Students exercising just 1 day per week had the highest headache rate (86.1%), while daily exercisers had the lowest (57.7%, *p* = 0.006). Regarding duration, exercising 31–59 min daily was linked to more headaches (91.0%) compared to ≤30 min (79.5%) or 1–2 h daily (68.8%, *p* = 0.001). Finally, 97.0% of students who reported exercise worsening their headaches experienced them in the past year versus 78.6% of those without exercise-induced headaches (*p* < 0.001).

Table 5 displays the results of the logistic regression analysis examining the factors associated with 12-month headache prevalence. In the univariate analysis, female gender, physical activities exacerbating headaches, using devices ≥4 h daily, and obesity were all significantly associated with headache prevalence. After adjusting for confounders in the multivariate model, headaches affecting daily activities remained strongly associated with increased odds of headaches (aOR 4.393, 95% CI 2.284–8.452, *p* < 0.001). Similarly, physical activities worsening headaches (aOR 10.37, 95% CI 2.02–53.35, *p* = 0.005) and using devices ≥4 h daily (aOR 13.89, 95% CI 1.96–98.54, *p* = 0.008) retained significant associations. Using devices 2–3 h daily also emerged as a risk factor in the multivariate model (aOR 6.34, 95% CI 0.80–50.36, *p* = 0.081).

## 4. Discussion

The 83% one-year prevalence of headaches among Jazan medical students is alarming, as it may impair the performance of these future healthcare professionals. The high prevalence likely results from several factors. For instance, third-year students showed the highest rate, at 88.3%, possibly due to demanding new clerkships provoking headaches through irregular sleep, meals, and increased stress [6]. 

Headache prevalence was also higher in females versus males (86.6% vs. 78.4%). Hormonal fluctuations during the menstrual cycle can trigger migraine headaches, which are more common in women [23]. Additionally, gender-specific psychosocial stressors may contribute to the sex differences [24].

Our findings align with studies demonstrating associations between electronic device overuse and headache disorders [10,11]. For instance, we found that students using devices ≥4 h daily had five-fold higher headache odds than those using devices <1 h daily. Frequent device users may experience visual strain, neck tension, and sleep disturbances, which can provoke headaches [25]. We also observed a negative correlation between physical activity and headaches, corroborating evidence that exercise helps prevent headaches [26]. Possible mechanisms include endorphin release, stress reduction, and improved sleep [27,28].

This study has limitations, including its cross-sectional design and self-reported data. Additionally, we did not examine other factors like diet, caffeine intake, or mental health. Strengths include the high response rate and comprehensive evaluation of multidimensional factors related to headaches. Overall, these findings highlight the urgent need for interventions, such as education campaigns, lifestyle training, and support services aimed at reducing excessive device use and inactivity to alleviate the burden of headaches among medical students. Longitudinal studies should continue to investigate potential mechanisms and targeted prevention strategies.

Our findings confirm the results of previous studies showing the associations between excessive electronic device usage and headache disorders, especially among student populations [2,11]. For instance, a study found that emergency medical students using devices for a long time daily had a higher headache risk than those with limited time of use [29]. Additionally, Abou Hashish et al., reported that increased electronic device time was significantly associated with more frequent headaches in Saudi health sciences students [8]. Together with our results, these findings provide evidence of headaches’ detrimental impacts on the quality of life, academic performance, and work productivity among those overusing electronic devices [13,15,30]. The potential mechanisms linking device overuse to headaches include visual strain, physical discomfort from prolonged static postures, and sleep disturbances from blue light exposure [11,31]. Our study contributes to the literature by demonstrating the scale of this issue among Saudi medical students, in particular, while highlighting opportunities for targeted interventions promoting responsible device habits to alleviate the headache burden in this group.

Furthermore, mounting evidence indicates that physical activity can prevent primary headaches. A systematic review by Varkey et al., found that regular exercise reduced migraine frequency, severity, and duration [26]. Our observation of an inverse headache–physical activity relationship provides further support for the claim that promoting exercise could alleviate headaches in medical students. The potential mechanisms include enhanced stress resilience and sleep quality [18]. Additionally, students lack knowledge about effective headache prevention like lifestyle modifications [14,16]. Therefore, multifaceted interventions providing education on trigger management, self-care techniques, and physical activity benefits, combined with support services, may improve headache outcomes. Overall, our study highlights the urgent need for evidence-based strategies focused on reducing electronic overuse and inactivity to mitigate the burden of primary headaches hampering medical students.

Several avenues remain open for future research to further elucidate the underlying mechanisms and effective management of primary headaches in medical students. Longitudinal studies should continue to explore the relationships between potential triggers like stress, sleep disturbances, dietary habits, and headache outcomes over time. Additionally, individual differences in age, gender, comorbidities, and other risk profiles should be examined to inform the development of personalized interventions. The long-term impacts of lifestyle modifications, including exercise, stress management, sleep hygiene, and nutrition, on headache patterns warrant investigation through intervention trials. Such studies could provide insight into optimal “doses” and combinations of non-pharmacological strategies to prevent and manage headaches. Cost-effectiveness analyses would also aid the translation of evidence-based approaches into large-scale implementation. Moreover, additional research applying strict ICHD diagnostic criteria is needed to accurately classify headache types and identify their underlying etiologies. We also recommend prospective longitudinal studies tracking headache patterns over time to differentiate between primary headaches like migraine versus secondary causes. Furthermore, future studies should directly compare the efficacy of different exercise modalities like aerobics, resistance training, and flexibility training for headache management [32]. Optimal exercise regimens tailored for headache disorders deserve exploration through randomized controlled trials. Combining exercise training with education, behavioral strategies, and the promotion of lifestyle change may provide additive benefits and improve adherence. Overall, elucidating the multifaceted biopsychosocial factors contributing to medical students’ headaches through robust longitudinal and experimental studies will be key to informing impactful, scalable, and sustainable interventions to enhance students’ health, well-being, and professional success.

### Limitations

This study has some limitations worth noting. The cross-sectional design prevents the determination of temporal relationships or causal inferences between headaches and associated factors. Additionally, the self-reported survey data could be vulnerable to recall bias in headache characteristics and social desirability bias in electronic device use or physical activity reporting. Our relatively small sample from a single institution may also limit the generalizability of findings to other populations. Furthermore, we did not evaluate other potentially relevant variables like stress, sleep quality, caffeine or medication use, and dietary triggers. We also did not apply ICHD criteria to definitively categorize headache types. Additional longitudinal studies with objective measures, larger representative samples, and assessments of multidimensional factors are warranted to corroborate and extend these findings. However, important strengths of our study include the high response rate and comprehensive evaluation of various headache dimensions, including patterns, disability, knowledge gaps, and modifiable lifestyle factors.

## 5. Conclusions

In summary, this cross-sectional study provides compelling evidence of a high burden of primary headaches impairing medical students at Jazan University. Excessive electronic device usage and physical inactivity emerged as potentially modifiable lifestyle factors associated with increased and decreased headache prevalence, respectively. These results underscore the urgent need for multifaceted interventions incorporating awareness campaigns, self-management training, and support services focused on the promotion of responsible device habits and regular exercise. Such initiatives may alleviate headaches and enhance productivity and well-being among medical students. However, longitudinal studies and trials are still needed to elucidate the mechanisms linking lifestyle factors and headaches over time. Evaluating stress, sleep, diet, and additional determinants will also help inform holistic, personalized, and sustainable approaches to prevent and treat headaches in this vulnerable population. Ultimately, evidence-based strategies that empower students to optimize self-care and effectively balance their demanding academic responsibilities will be key to mitigating this public health issue threatening the next generation of healthcare professionals.

## Figures and Tables

**Figure 1 medicina-60-00299-f001:**
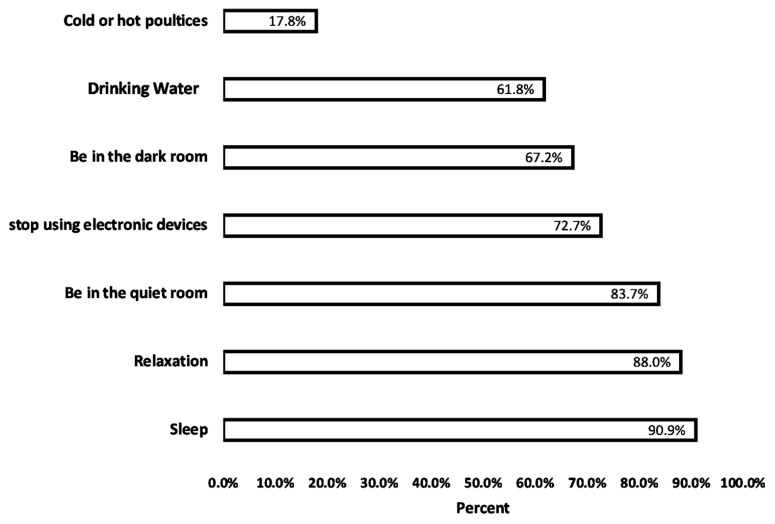
Different methods to cope with primary headaches reported by study participants.

**Table 1 medicina-60-00299-t001:** Sociodemographic characteristics and 1-year prevalence of headaches classified by selected characteristics.

Characteristics		All Participants		Prevalence during the Past 12 Months				
		*N*	%	*N*	%	95% CI		*p* Value
						Lower	Upper	
Gender	Male	273	(54.2)	214	78.4%	73.2%	83.0%	0.017
	Female	231	(45.8)	200	86.6%	81.7%	90.5%	
Age Groups (years)	18–21	223	(44.2)	189	84.8%	79.6%	89.0%	0.372
	22–24	211	(41.9)	168	79.6%	73.8%	84.6%	
	25–30	70	(13.9)	57	81.4%	71.2%	89.2%	
Marital Status	Single	486	(96.4)	399	82.1%	78.5%	85.3%	0.893
	Married	18	(3.6)	15	83.3%	61.9%	95.1%	
Academic Year	2nd year	97	(19.2)	80	82.5%	74.0%	89.0%	0.385
	3rd year	111	(22.0)	98	88.3%	81.3%	93.3%	
	4th year	122	(24.2)	100	82.0%	74.4%	88.0%	
	5th year	66	(13.1)	52	78.8%	67.8%	87.3%	
	6th year	69	(13.7)	55	79.7%	69.1%	87.9%	
	Medical Intern	39	(7.7)	29	74.4%	59.3%	86.0%	
CGPA	2:00–3:00	16	(3.2)	16	100.0%	-	-	0.112
	3.01–3.50	53	(10.5)	44	83.0%	71.3%	91.3%	
	3.51–4.00	85	(16.9)	70	82.4%	73.2%	89.3%	
	4.01–4.50	114	(22.6)	86	75.4%	67.0%	82.6%	
	4.51–5.00	236	(46.8)	198	83.9%	78.8%	88.2%	
Father’s Education	Basic Education *	61	(12.1)	45	73.8%	61.8%	83.5%	0.104
	High School Education	81	(16.1)	70	86.4%	77.7%	92.6%	
	Undergraduate Education	314	(62.3)	263	83.8%	79.4%	87.5%	
	Graduate Education	48	(9.5)	36	75.0%	61.5%	85.5%	
Mother’s Education	Basic Education	135	(26.8)	115	85.2%	78.5%	90.4%	0.025
	High School Education	84	(16.7)	62	73.8%	63.7%	82.3%	
	Undergraduate Education	272	(54.0)	229	84.2%	79.5%	88.2%	
	Graduate Education	13	(2.6)	8	61.5%	35.0%	83.5%	
BMI Categories	Underweight	82	(16.3)	64	78.0%	68.2%	85.9%	0.571
	Normal Weight	237	(47.0)	193	81.4%	76.1%	86.0%	
	Overweight	115	(22.8)	97	84.3%	76.9%	90.1%	
	Obese	70	(13.9)	60	85.7%	76.1%	92.4%	

Abbreviations: CGPA = cumulative grade points average; BMI = body mass index; CI = confidence interval; * *p* value is based on chi-square test. Basic education refers to less than secondary education.

**Table 2 medicina-60-00299-t002:** Patterns and characteristics of headaches among study participants according to gender.

Characteristics		All Participants		Gender				
				Male		Female		*p* Value
		*N*	%	*N*	%	*N*	%	
During the last 30 days, how many days did you have a headache?	Just one day	106	(24.6)	65	(28.9)	41	(19.9)	0.001
	2–5 days	190	(44.1)	102	(45.3)	88	(42.7)	
	6–10 days	90	(20.9)	47	(20.9)	43	(20.9)	
	11–20 day	33	(7.7)	9	(4.0)	24	(11.7)	
	21–30 day	12	(2.8)	2	(0.9)	10	(4.9)	
How long does this headache usually last?	Less than 30 min	95	(22.1)	52	(23.1)	43	(21.1)	0.103
	31 min to one hour	102	(23.8)	58	(25.8)	44	(21.6)	
	1–2 h	81	(18.9)	50	(22.2)	31	(15.2)	
	2–3 h	61	(14.2)	26	(11.6)	35	(17.2)	
	3–4 h	34	(7.9)	15	(6.7)	19	(9.3)	
	More than 4 h	56	(13.1)	24	(10.7)	32	(15.7)	
How many pills do you take to treat the headache on one occasion?	I don’t take any pills	206	(48.1)	115	(51.6)	91	(44.4)	0.500
	1–2	207	(48.4)	101	(45.3)	106	(51.7)	
	3–4	10	(2.3)	5	(2.2)	5	(2.4)	
	More than 4 pills	5	(1.2)	2	(0.9)	3	(1.5)	
How many days in the last 30 days did you take these medications?	I don’t take any pills	182	(42.8)	107	(48.2)	75	(36.9)	0.016
	Just one day	106	(24.9)	47	(21.2)	59	(29.1)	
	2–5 days	98	(23.1)	55	(24.8)	43	(21.2)	
	6–10 days	24	(5.6)	7	min (3.2)	17	(8.4)	
	11–20 days	13	(3.1)	6	(2.7)	7	(3.4)	
	21–30 days	2	(0.5)	0	(0.0)	2	(1.0)	
How long would it last if you did not take medication?	1 = I don’t take any pills	143	(33.6)	84	(37.8)	59	(28.9)	0.014
	Less than 30 min	32	(7.5)	15	(6.8)	17	(8.3)	
	31 min-1 h	48	(11.3)	28	(12.6)	20	(9.8)	
	1–2 h	54	(12.7)	34	(15.3)	20	(9.8)	
	2–3 h	47	(11.0)	23	(10.4)	24	(11.8)	
	3–4 h	23	(5.4)	7	(3.2)	16	(7.8)	
	More than 4 h	79	(18.5)	31	(14.0)	48	(23.5)	
Headache Pain Level	Minor	86	(20.2)	46	(20.6)	40	(19.7)	0.906
	Moderate	239	(56.1)	126	(56.5)	113	(55.7)	
	Severe	101	(23.7)	51	(22.9)	50	(24.6)	
Where usually do you feel the headache in your head?	In left or right side	87	(20.4)	48	(21.5)	39	(19.1)	
	In both sides	98	(23.0)	39	(17.5)	59	(28.9)	0.048
	Forehead	182	(42.6)	102	(45.7)	80	(39.2)	
	Back of the head	60	(14.1)	34	(15.2)	26	(12.7)	
Thinking still of this type of headache, which best describes the pain?	Throbbing or pulsating	92	(21.6)	59	(26.6)	33	(16.2)	0.012
	Pressing or squeezing	175	(41.1)	89	(40.1)	86	(42.2)	
	Sharp or stabbing	39	(9.2)	14	(6.3)	25	(12.3)	
	Every attack with different pattern	120	(28.2)	60	(27.0)	60	(29.4)	
What are the things that trigger your headache and worsen it?	Exercises	32	(7.5)	22	(10.0)	10	(4.9)	0.019
	Light exposure	157	(36.9)	68	(30.8)	89	(43.6)	
	Noise	138	(32.5)	79	(35.7)	59	(28.9)	
	Other	98	(23.1)	52	(23.5)	46	(22.5)	
With this type of headache, do you usually feel nauseated	Yes	88	(20.8)	27	(12.2)	61	(30.2)	0.001
	No	335	(79.2)	194	(87.8)	141	(69.8)	
With this type of headache, do you usually actually vomit	Yes	22	(5.2)	7	(3.2)	15	(7.4)	0.049
	No	401	(94.8)	214	(96.8)	187	(92.6)	
Has a healthcare professional ever diagnosed you with any headache?	Yes	21	(4.2)	12	(4.4)	9	(3.9)	0.708
	No	483	(95.8)	261	(95.6)	222	(96.1)	
Type of headache have you been diagnosed with	Migraine	20	(33.3)	12	(35.3)	8	(30.8)	0.106
	Tension	8	(13.3)	4	(11.8)	4	(15.4)	
	Cluster	4	(6.7)	0	(0.0)	4	(15.4)	
	I don’t remember	28	(46.7)	18	(52.9)	10	(38.5)	
Seeking medical advice	Yes	29	(22.0)	21	(31.8)	8	(12.1)	0.006
	No	103	(78.0)	45	(68.2)	58	(87.9)	
Do you think they are all of one type, or are they of more than one type?	One type	43	(30.7)	23	(32.4)	20	(29.0)	0.803
	More than one time	59	(42.1)	28	(39.4)	31	(44.9)	
	I’m not sure	38	(27.1)	20	(28.2)	18	(26.1)	

**Table 3 medicina-60-00299-t003:** Electronic device use and its association with primary headache.

Characteristics		Headache during the Last 12 Months?				*p* Value
		Yes		No		
		*N*	%	*N*	%	
Do you use electron devices	Yes	410	(82.8)	85	(17.2)	0.003
	No	4	(44.4)	5	(55.6)	
What electron devices do you use most during the day?	Television	3	(75.0)	1	(25.0)	0.165
	Smartphone	183	(78.2)	51	(21.8)	
	Tablet device (Tablet or iPad)	209	(86.0)	34	(14.0)	
	Computer	17	(81.0)	4	(19.0)	
How many hours do you use your primary device * during the day?	Less than 30 min	4	(57.1)	3	(42.9)	0.006
	31 min–1 h	2	(40.0)	3	(60.0)	
	1–2 h	13	(61.9)	8	(38.1)	
	2–3 h	23	(85.2)	4	(14.8)	
	3–4 h	40	(87.0)	6	(13.0)	
	More than 4 h	332	(83.4)	66	(16.6)	
How many hours do you use all your electronic devices ** during the day?	Less than 30 min	2	(50.0)	2	(50.0)	0.173
	31 min–1 h	5	(83.3)	1	(16.7)	
	1–2 h	6	(60.0)	4	(40.0)	
	2–3 h	10	(66.7)	5	(33.3)	
	3–4 h	27	(77.1)	8	(22.9)	
	More than 4 h	364	(83.9)	70	(16.1)	
How is the rate of your use of electronic devices during one time?	Continuously without stopping	278	(84.0)	53	(16.0)	0.135
	Interrupted	136	(78.6)	37	(21.4)	

* Refers to the device type used most during the day by each participant; ** indicates the total combined usage time across all devices.

**Table 4 medicina-60-00299-t004:** Association between headache and physical activity.

Characteristic		Have You Had a Headache during the Last 12 Months?				*p* Value
		Yes		No		
		*N*	%	*N*	%	
Do you exercise or do any physical activities in your day?	Yes	169	(79.7)	43	(20.3)	0.226
	No	245	(83.9)	47	(16.1)	
Do you exercise regularly?	Yes	62	(74.7)	21	(25.3)	0.053
	No	352	(83.6)	69	(16.4)	
How much do you exercise per week?	One day	130	(86.1)	21	(13.9)	0.006
	2 to 3 days	91	(82.7)	19	(17.3)	
	4 to 6 days	28	(77.8)	8	(22.2)	
	Every day	15	(57.7)	11	(42.3)	
How long do you exercise per day?	30 min or less	128	(79.5)	33	(20.5)	0.001
	31 min to 59 min	101	(91.0)	10	(9.0)	
	1–2 h	22	(68.8)	10	(31.3)	
	More than 2 h	9	(75.0)	3	(25.0)	
Does physical activities or exercise tend to make the headache worse?	Yes	64	(97.0)	2	(3.0)	<0.001
	No	242	(78.6)	66	(21.4)	

**Table 5 medicina-60-00299-t005:** Independent predictors of the factors associated with primary headaches using logistic regression analyses.

Characteristics		Univariate				Multivariate			
		cOR	95% CI		*p* Value	aOR	95% CI		*p* Value
			Lower	Upper			Lower	Upper	
Gender	Male *	1				1			
	Female	1.78	1.11	2.86	0.018	1.58	0.83	3.01	0.168
Daily activities affected by headache	No *	1							
	Yes	4.08	2.31	7.19	<0.001	4.393	2.284	8.452	<0.001
physical activities affecting severity of headaches	No *	1				1			
	Yes	8.73	2.08	36.60	0.003	10.37	2.02	53.35	0.005
Electronic device use	<one hour *	1				1			
	2–3 h	3.00	0.81	11.08	0.099	6.34	0.80	50.36	0.081
	4 or more hours	5.03	1.57	16.08	0.006	13.89	1.96	98.54	0.008
BMI categories	Underweight *	1				1			
	Normal weight	1.23	0.67	2.29	0.505	1.43	0.61	3.35	0.412
	Overweight	1.52	0.73	3.13	0.261	1.89	0.72	4.95	0.193
	Obese	1.69	0.72	3.95	0.227	3.25	0.93	11.43	0.066

* Reference category; CI = confidence interval; cOR = crude odds ratio; aOR = adjusted odds ratio.

## Data Availability

The data that support the findings of this study are also available from the corresponding author upon reasonable request.

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
