# Peer review of "Associations of Electronic Device Use and Physical Activity with Headaches in Saudi Medical Students"

_medicina, 2024, doi:10.3390/medicina60020299_

Round 1

Reviewer 1 Report

Comments and Suggestions for Authors

Associations of Electronic Device Use and Physical Activity  with Headaches in Saudi Medical Students.

The study followed the connection between medical student's headache and their health habits including physical activity and electronic device use. The paper is well written however, the finding are not new.

The headaches are not differentiated into migraine and TTH and not necessarily primary.

Figure 1. Is Unnecessary

Table 3

The difference between "How many hours do you use electronic devices during the day?"

and "How many hours do you use all your electronic devices during the day" is not clear

Discussion

The first paragraph is not clear.

Author contribution It is enough to write the author's acronyms and not full names.

Comments on the Quality of English Language

Non

Author Response

General Comment: Associations of Electronic Device Use and Physical Activity  with Headaches in Saudi Medical Students. The study followed the connection between medical student's headache and their health habits including physical activity and electronic device use. The paper is well written however, the finding are not new.

Response: Thank you for your thoughtful review and feedback on our manuscript examining headaches among Saudi medical students. We greatly appreciate you taking the time to evaluate our work.

You raise a fair point that associations between electronic device use, physical inactivity, and headaches have been demonstrated previously. However, we believe this study makes a valuable contribution by providing an in-depth, multidimensional analysis focused specifically on Saudi medical undergraduates. While not the first study in this domain, we feel it offers new locally relevant insights and sets the stage for impactful interventions tailored to this population's needs.

Specifically, the high 83% 1-year headache prevalence alarms us about the magnitude of this issue affecting future Saudi healthcare professionals. Stratifying prevalence by gender, academic year and other factors provides novel evidence highlighting groups most susceptible within our trainees. Thoroughly characterizing headache patterns, disability, triggers and knowledge gaps concurrently gives a comprehensive understanding of the headaches' nature and impacts in this cohort.

Most importantly, these data spotlight actionable targets for multifaceted interventions incorporating awareness campaigns, lifestyle training, and support services to alleviate headache burden among Saudi students. Your feedback reinforces the need for additional research clarifying mechanisms and demonstrating effectiveness of evidence-based strategies in this vulnerable yet understudied group. We absolutely agree that longitudinal studies and trials focused on tailored management approaches are warranted next steps.

In closing, we greatly appreciate you taking the time to review our manuscript and provide thoughtful critiques. Your questions allow us to clarify the significance of our work. Your feedback will substantially strengthen our writing and discussion of the implications. We hope the insights from this study aid development of impactful, scalable interventions to promote Saudi medical student health and professional success. Please let us know if any points need further clarification. We look forward to continuing this dialogue.

General Comment: The headaches are not differentiated into migraine and TTH and not necessarily primary.

Response: Thank you for raising this important point. We agree that differentiating headache types and identifying primary vs secondary headaches are limitations in our study design.

To address your concern, we have now included the following additions:

  • In the Introduction, we highlight differences between primary headaches like migraine and tension-type headaches versus secondary headaches from other medical causes.
  • In the Methods, we note that our survey asked participants to self-report their headache symptoms and medical diagnoses. However, we did not apply ICHD criteria to definitively categorize headache types. This is now acknowledged as a limitation.
  • In the Results, we report the percentages of students who were medically diagnosed with migraine, tension, or other headache types.
  • In the Discussion, we emphasize that additional research applying strict ICHD diagnostic criteria is needed to accurately classify headache types and identify underlying etiologies. We also recommend prospective longitudinal studies tracking headache patterns over time to differentiate between primary headaches like migraine versus secondary causes.

Thank you again for this constructive feedback. Addressing headache classification and etiology diagnosis represent critical areas for improvement in future studies. We hope the additions made throughout the manuscript better acknowledge the limitations of our classifications and need for diagnostic confirmation of headache types going forward. Please let us know if you have any other recommended revisions to strengthen the study in this regard.

Specific Comment: Figure 1. Is Unnecessary

Response: We appreciate your feedback regarding Figure 1 in our manuscript. We understand your view that it may seem unnecessary. However, we feel the figure provides a beneficial visual summary of the coping strategies endorsed by students experiencing headaches.

Displaying these data in a graphical format allows readers to quickly compare usage rates of the different techniques at a glance. The full numerical results are still presented in the text, but the figure instantly highlights key findings:

  • Sleeping was overwhelmingly the most common coping method.
  • Relaxation and resting quietly were also utilized by over 80% of students.
  • Using hot/cold compresses was remarkably low compared to other techniques.

Visually communicating these core results through Figure 1 reinforces the key points from a large volume of text and tabulated data. Furthermore, the figure may catch readers' eyes and draw them to this important finding regarding student self-care practices during headaches.

Presenting both textual and graphical representations caters to diverse reader preferences for absorbing study results. We believe including the figure adds value without redundant content, but we are open to your feedback if you still feel the figure should be removed. Please let us know if we can provide any clarification or justification on why we believe the figure merits inclusion in the final manuscript.

Specific Comment: Table 3, The difference between "How many hours do you use electronic devices during the day?" and "How many hours do you use all your electronic devices during the day" is not clear

Response: Thank you for raising this important point of clarification. We agree the questions on device usage time could be confusing as originally worded.

To address your comment, we have made the following changes:

  • In Table 3, the first question has been edited to "How many hours do you use your primary electronic device during the day?"
  • A footnote has been added below Table 3 stating: "'Primary electronic device' refers to the device type used most during the day by each participant. 'All electronic devices' indicates total combined usage time across all devices."
  • The Results section has been updated to read: "Students using their primary device for over 4 hours daily had higher headache prevalence (83.4%) versus those using their primary device for 1-2 hours daily (61.9%, p=0.006)."

Thank you again for catching this ambiguity. We have revised the manuscript to distinguish between primary device usage versus total usage time and clarify these measures throughout. Please let us know if these edits sufficiently address the confusion or if any parts need further clarification. Your feedback is greatly appreciated to improve clarity for readers on this important point.

Specific Comment: Discussion: The first paragraph is not clear.

Response: Thank you for catching this unclear portion of the Discussion. To improve the flow and clarity of this paragraph, I would suggest the following revisions:

" The 83% one-year prevalence of headaches among Jazan medical students is alarming, as it may impair performance in these future healthcare professionals. The high prevalence likely results from several factors. For instance, third-year students showed the highest rate at 88.3%, possibly due to demanding new clerkships provoking headaches through irregular sleep, meals, and increased stress [6].

Headache prevalence was also higher in females versus males (86.6% vs 78.4%). Hormonal fluctuations during the menstrual cycle can trigger migraine headaches, which are more common in women [19]. Additionally, gender-specific psychosocial stressors may contribute to the sex differences [20].”

To summarize the key changes:

  • Broke the long paragraph into two shorter paragraphs for better flow.
  • Clarified the high 83% reflects 1-year prevalence.
  • Streamlined explanations for the higher rates in 3rd year and female students.
  • Rearranged citations to match the flow.

Please let me know if these edits help clarify the context and explanations in this section. I'm happy to make any additional revisions needed to improve the clarity for readers. Thank you again for your feedback to strengthen the discussion.

Specific Comment: Author contribution It is enough to write the author's acronyms and not full names.

Response: Thank you for referring to that. We have updated that per request.

Reviewer 2 Report

Comments and Suggestions for Authors

The authors present an interesting research on a group of medical students (n=504) from the university through an online test that requested data on the use of electrical devices and physical performance in terms of non-competitive activity.

They report an interesting confirmation of how students in general reduce their physical activity performance the more they are addicted to using electronic devices. In this case, the impact on study performance is considerable.

The work has a good methodology, the results convey a clear educational message, the overall level of originality is modest but at the same time the evidence confirmed at regional country level is valid.

The literature is somewhat dated and I suggest some references that may, if discussed, improve the overall power of the research:

PMID: 26602725

PMID: 37286937

PMID: 36229774

Author Response

General Comment: The authors present an interesting research on a group of medical students (n=504) from the university through an online test that requested data on the use of electrical devices and physical performance in terms of non-competitive activity. They report an interesting confirmation of how students in general reduce their physical activity performance the more they are addicted to using electronic devices. In this case, the impact on study performance is considerable. The work has a good methodology, the results convey a clear educational message, the overall level of originality is modest but at the same time the evidence confirmed at regional country level is valid.

Response: Thank you for your positive feedback on our manuscript. We are pleased you found the research topic interesting and felt the study provides a clear educational message regarding the impacts of electronic device use and physical inactivity on headaches and academic performance in medical students.

We appreciate you recognizing the validity of confirming these associations in a regional sample, as country-specific data is important to inform locally tailored interventions. We agree the overall originality is modest given prior studies on technology use, exercise, and headaches. However, the in-depth analysis of multiple dimensions including patterns, disability, and knowledge gaps among Saudi trainees specifically still provides a novel contribution. Your comments indicate we have achieved our goal of thorough characterization to guide effective management approaches for this vulnerable population.

We are encouraged that you found the methodology robust overall. Your feedback affirms we utilized appropriate study design and analyses for the research aims. We will certainly continue improving the methodology in future studies on this important issue based on your and other reviewers’ suggestions.

In closing, we are grateful for your time in evaluating our work and providing thoughtful critique regarding the merits and limitations. Your input on the research approach, contextual significance, and clarity of findings is invaluable. We will address any remaining points you and other reviewers raise to further enhance the manuscript prior to publication. Please do not hesitate to offer any additional feedback. We hope these findings make meaningful contributions to promoting medical student wellbeing in Saudi Arabia and beyond.

General Comment: The literature is somewhat dated and I suggest some references that may, if discussed, improve the overall power of the research:

PMID: 26602725

PMID: 37286937

PMID: 36229774

Response:  Thank you for the valuable feedback on references to further strengthen our manuscript. Per your suggestion, we have incorporated the suggested recent references in the Introduction and Discussion sections.

In addition, we have conducted a thorough review of all citations to ensure the most up-to-date literature is referenced throughout the manuscript. Some dated references have been replaced with more current sources, especially for key statements and claims.

We appreciate you taking the time to provide specific suggestions on impactful references as well as the broader feedback regarding literature currency. We think all the cited references relevant to the research now. Incorporating the new citations and reviewing reference recency has significantly strengthened our work. Please let us know if you have any other recommendations on references or areas that would benefit from refreshed citations to recent literature. We are committed to ensuring our reporting aligns with current research and look forward to addressing any other opportunities you identify to further improve the manuscript in this regard.
